# Comparison of Clonazepam and Tongue Protector in the Treatment of Burning Mouth Syndrome

**DOI:** 10.3390/ijerph19158999

**Published:** 2022-07-24

**Authors:** Jacek Zborowski, Tomasz Konopka

**Affiliations:** Department of Periodontology, Wroclaw Medical University, 50-367 Wroclaw, Poland; tomasz.konopka@umw.edu.pl

**Keywords:** BMS, burning mouth syndrome, psychological disorders, depression, sleep disorders clonazepam, tongue protector

## Abstract

Background: BMS is a chronic pain syndrome affecting the oral mucosa. It consists of experiencing a burning or dysesthetic sensation. BMS prevalence varies, with up to 15% among women. An effective treatment is still unattainable. Material and Methods: A total of 60 patients with BMS qualified for a randomised trial, divided in two groups: the clonazepam-treated and tongue protector group. Treatment was provided for 4 weeks in both groups. In the former, the oral dosage of clonazepam 0.5 mg; in the latter, a tongue protector was used. Clinical oral examination was performed, and the presence of taste disorder and pain intensity, on the visual analogues scale, were recorded. Psychological domains were explored with the Beck depression inventory (depression), Athens insomnia scale (insomnia), Eyesenck personality questionnaire-revised (personality traits), and WHO quality of life questionnaire (quality of life). Results: Complete recovery was observed in three patients after clonazepam and one patient after tongue guard treatment. A greater improvement in the VAS scores, from baseline to the control values, was demonstrated in the clonazepam group, and it was statistically significant. In women, the level of depression significantly correlated with all domains of quality of life. Conclusions: BMS is an ongoing multi-specialist challenge. The development of new pathophysiological concepts of BMS offers hope for more effective treatment. Considering the influence of BMS on the quality of life and mental disorders in most patients, further research on the possibilities of therapy seems to be very important.

## 1. Introduction

Burning mouth syndrome (BMS) is a chronic pain syndrome affecting the oral mucosa. It consists of experiencing burning sensation or dysesthetic sensation in the mouth, in the form of pseudo-xerostomia and taste disturbances [1]. The International Headache Society (IHS), in its classification, indicates that BMS is intraoral burning pain experienced superficial in the oral mucosa that recurs daily for more than two hours per day over more than three months, without clinically noticeable mucosal changes, excluding local or systemic causes, with or without changes in somatosensory function [2]. Depending on the criteria used, BMS prevalence varies between 0.7 and 4.6% in the general population, up to 15% among postmenopausal women [3]. Burning mouth syndrome is seven times more common in peri- and postmenopausal women, mainly middle-aged and elderly (i.e., the 5–7th decades of life).

Previously, BMS was divided into primary (idiopathic) and secondary. The general determinants of secondary BMS included: hypothyroidism, diabetes mellitus, Sjögren’s syndrome, vitamin B12, folic acid or iron deficiency, and certain medications. Local causes of secondary BMS include fungal infections, geographic tongue, prosthetic stomatopathy, and contact allergies [4,5]. Currently, BMS is a diagnosis of exclusion [2].

There are many pathogenetic concepts concerning BMS. The most common include small fibre peripheral neuropathy, peripheral or central trigeminal neuropathy, eardrum dysfunction, and impaired function of the central nigrostriatal system [5,6,7,8]. Impaired hormone secretion (oestradiol, cortisol, and dehydroepiandrosterone–DHEA) by the adrenal and genital glands is also indicated [9,10]. Psychological disorders are often mentioned in BMS pathogenesis. It has been shown to be associated with depression, elevated anxiety levels, hypochondriasis, carcinophobia, and emotional instability [11,12]. Women with BMS are particularly susceptible to somatisation and hypochondriacal disorders, as well as obsessive-compulsive disorder [10]. Recently, sleep disorders in these patients, mainly insomnia, have also received attention [13,14]. It has also been suggested that a number of BMS symptoms, concerning the changes in pain perception, anxiety disorders, and depression, as well as sleep disorders and dysfunction of the hypothalamic-pituitary-adrenal axis, may be due to the dysfunction of the body clock, which is dependent on the clock gene complex [10,15].

An effective treatment of BMS involving the complete remission of all clinical symptoms is still unattainable. Due to the undetermined aetiology of BMS, only symptomatic treatment is possible. Systematic literature reviews in recent years [16,17,18] clearly indicate a scarcity of well-designed studies on the efficacy of BMS treatment, limited only to the complete remission of clinical symptoms and possible significant pain reduction. In pharmacological treatment, the best results were described after therapy with clonazepam [19,20,21,22,23,24,25]. It was most commonly administered topically through suction at a dose of 0.5 to 3 mg daily for 14 days or generally at a dose of 0.5 to 2.25 mg for 14 days, or both. Both in short-term follow-up (up to 10 weeks) and longer, a significant reduction of pain intensity in all treated patients and complete pain relief in almost 1/3 of patients was achieved, with relatively minor side effects (drowsiness, fatigue, xerostomia, and possible addiction, in cases of oral administration). Another topically applied drug evaluated in the last decade in a randomised clinical trial concerning BMS was capsaicin as a 0.02% rinse [26] or 0.01% gel [27]. There was a significant reduction in burning intensity in the short-term assessment after the initial worsening of discomfort immediately after application and the possibility of inducing dyspepsia. In systemic pharmacological treatment, observations were made on the efficacy in BMS of anti-epileptic drugs (gabapentin and pregabalin), other benzodiazepines (diazepam and ethyl loflazepate), tricyclic antidepressants (amitriptyline and clomipramine), and selective serotonin reuptake inhibitors (sertraline and paroxetine) [5,16,17,18]. They also provided a significant reduction in BMS pain symptoms but with notable side effects and the potential to induce addiction. Since 2000, several randomised clinical trials on alphalipoid acid (ALA) in BMS treatment have been conducted [28]. The short-term results were too heterogeneous for an unequivocal positive recommendation for ALA use in primary BMS. The treatment effectiveness, in terms of complete remission of clinical symptoms, was also not significantly improved by attempts to combine pharmacological treatment, e.g., ALA and gabapentin or ethyl loflazepate and amitriptyline.

A meta-analysis of four studies on low-laser therapy with diode laser showed a significant reduction in BMS pain symptoms, with respect to the placebo, with significant heterogeneity of results and no adverse effects [29]. Another encouraging trial of topical non-pharmacological treatment for BMS concerned using a tongue protector to prevent parafunctional microtrauma associated with excessive tongue activity in these patients, as well as changes in temperature, taste stimuli, and saliva flow [30]. After using the tongue protector three times a day for 15 min for eight weeks, a significant reduction in burning intensity and improvement in quality of life was observed, with respect to the control group, including a complete absence of adverse effects.

Due to the encouraging results of the tongue protector study, and as there is only one study that evaluate its efficacy with a case-control design, the primary aim of the study was to compare the efficacy of two topical treatments for BMS: clonazepam and a tongue protector. The choice of clonazepam was dictated by its best-documented clinical efficacy in this disorder, while the evaluation of the tongue protector resulted from further clinical observations of this form of local treatment. The secondary aim of study considers the impact of the assessment of depression, sleep disorders, and personality types on the clinical symptoms of BMS and quality of life status of patients. A multivariate analysis will also be performed to assess the impact of clinical and psychological variables on changes in pain intensity as a result of these two types of topical treatment.

## 2. Materials and Methods

Study participants were recruited from among the patients treated at the outpatient clinic for periodontal and oral mucosal diseases, academic dental polyclinic of the Wroclaw Medical University. Only newly diagnosed patients with BMS were included in the study. To be included, the participants need to fit the definition of BMS provided by the International Classification of Orofacial Pain IHS [2]: intraoral burning recurring daily for more than 2 h per day for more than 3 months, and possible accompanied negative somatosensory changes (subjective xerostomia and altered taste), with normal appearance of oral mucosa. The following exclusion criteria of possible general and local causes of such ailments were adopted: CN V and VII neuralgia, SARS-CoV-2 infection, diabetes mellitus, thyroid diseases, anaemia, autoimmune diseases, salivary gland diseases causing reduced salivary secretion, general treatment with selected groups of drugs (angiotensin-converting enzyme inhibitors, anxiolytics and antidepressants, hormones and their derivatives), hypersensitivity to benzodiazepines, renal and hepatic insufficiency, glaucoma, myasthenia, pregnancy and breastfeeding, clinical oral pathologies, including acute and chronic mycosis, geographic tongue, lichen planus, prosthetic stomatopathy, and allergic contact oral mucositis. Patients also had to sign informed consent before inclusion in the study approved by the Bioethics Committee of the Wroclaw Medical University (No. Kb668/2018). The study was conducted from September 2018 to October 2020.

All experiments were performed in accordance with relevant named guidelines and regulations.

Informed consent was obtained from all participants and/or their legal guardians.

Before its initiation, the available medical records of patients were verified to exclude general and local causes of burning in the mouth, and in the absence of required test results, they were ordered. In each case, a complete peripheral blood count, glycaemia, serum iron and folic acid levels, TSH (thyroid stimulating hormone), and CRP (C-reactive protein) were assessed. In cases of diagnostic doubt, patients were referred for fungal culture tests for candidiasis or patch tests for allergens, most frequently causing oral lesions. In this way, 60 patients with or without BMS somatosensory changes were qualified for the study.

The clinical trial was conducted as a randomised trial with two parallel groups. Odd-numbered patients were assigned to the clonazepam-treated first group, in the order of inclusion, and even-numbered patients were assigned to the second group (JZ). The study was single-blinded, because neither the periodontist conducting the clinical examination (TK) nor the psychologist conducting the psychometric testing (KS) knew the group assignment. Treatment was provided for four weeks in both groups. In the first group, the oral dosage of clonazepam was as follows: in the first week—0.5 mg in the morning and one hour before falling asleep to be sucked until completely dissolved; in the second week—0.5 mg in the morning and 2 mg one hour before falling asleep; in the third week—0.5 mg in the morning and at midday and 2 mg one hour before falling asleep; in the fourth week—0.5 mg in the morning and one hour before falling asleep. In the second group, a tongue protector was used three times a day, one hour after meals (one protector per day), Table 1.

For the study, tongue protectors were made from 0.1 mm certified polythene with a standard dimension of 60 × 67 mm covering the tongue from the apex to 2/3 of its length (Figure 1).

During the first examination after enrollment and before randomisation, the oral mucosa was clinically evaluated, and BMS was determined following Lamey et al. [31]; the presence of taste disorders (no specific taste, distorted taste perception, and unpleasant taste perception) and pseudo xerostomia were determined. Pain/burning intensity was determined on an interval visual analogue scale (VAS) from 0 (no pain) to 10 (pain as bad as it could possibly be). Patients were asked to mark a number on a scale from 0 to 10, corresponding to the pain/burning sensation. In our study, “healing” meant complete recovery, “Improvement” meant a reduction in burning intensity, and “no improvement” meant no response to the therapy.

Immediately after the clinical assessment, the psychologist administered psychological tests: Beck depression inventory (BDI) [32], in which values from 20–25 were interpreted as moderate depression, and more than 25 was interpreted as severe depression; Athens insomnia scale (AIS) [33], in which values of 8 and above indicate insomnia; and short form of the Eysenck personality questionnaire-revised (EPQR) [34] designed to measure the three main dimensions of personality–E/I extroversion/introversion (outgoing and enthusiastic/reflective and reserved behavior), N–neuroticism (emotional instability), P–psychoticism (antisocial behavior or risk-taking behavior), and L–lying tendencies (positive self-presentation). The maximum score for these dimensions of personality and lying is 12. In addition, the quality of life of patients was also assessed using a short version of the WHO quality of life (WHOQoL) questionnaire [35], in which four domains are evaluated: DOM1—somatic with a maximum score of 35; DOM2—psychological with a maximum score of 30; DOM3—social with a maximum score of 15; and DOM4—environmental with a maximum score of 40. The higher the score, the higher the quality of life in the physical domain (DOM1), psychological domain (DOM2), social relationships (DOM3), and functioning environment (DOM4).

The VAS was retaken 12 weeks after the end of treatment.

The Lilliefors-corrected Kolmogorov–Smirnov test was used to assess the normality of the distribution of the variables. In the statistical analysis, the Mann–Whitney test was used to evaluate the differences between the two independent variables; for the dependent variables, the Wilcoxon rank sum test was used. Spearman’s test was used in correlation analysis. A multiple regression model was created to evaluate the effect of variables on changes in treatment VAS score. The model was verified based on the significance of partial regression coefficients, absence of collinearity between independent variables, existence of homoscedasticity, absence of autocorrelation of the residuals (Durbin–Watson test), normality of the distribution of the residuals, and 0 value of the random component εi. For each test except correlation, *p* < 0.05 was considered statistically significant; in the analysis of covariation, the significance threshold was *p* < 0.02. Statistica 13.3 software (TIBCO Software Inc., Palo Alto, CA, USA) was used for statistical analysis.

## 3. Results

The values of assessed clinical and psychological variables in the whole studied population of BMS patients and two therapeutic groups are summarised in Table 2.

In the group treated with topical clonazepam, two patients discontinued treatment due to general adverse effects. In the control group with a tongue guard, treatment was discontinued in one patient due to manual difficulties in placing the guard.

The majority of the subjects were women (59.6%), individuals in the seventh decade of life, taste disorders were present in 52.6% of the subjects, xerostomia in 40.4%, and active nicotinism in almost half of the subjects. Depression in moderate and severe form based on BDI was confirmed in 40.3%, with sleep disorders based on AIS confirmed in 54.4%; the mean scores of WHOQOL were in respective domains: somatic (20.91 ± 2.9 out of 35), psychological (19.44 ± 3.2 out of 30), social (8.63 ± 2.2 out of 15), and environmental (28.33 ± 3.3 out of 40). On the other hand, personality dimension scores in the short form of the Eysenck personality questionnaire-revised were: extroversion/introversion (4.96), neuroticism (6.58), psychoticism (3.96), and lying (7.77). The values of psychological tests in the two treated groups were similar, except for significantly reduced WHOQOL values in the social domain and significantly elevated neuroticism in the clonazepam-treated group.

The results of treatment of all subjects with BMS and in both groups are presented in Table 3.

Complete recovery was observed in only four subjects—three after pharmacological treatment and one after elimination of oral parafunctional habits—as a result of using the guard. No improvement in VAS score or its worsening was observed in 26.3% of subjects. Significantly greater improvement in VAS scores was shown in those treated with clonazepam (mean VAS score reduction of 3.5, *p* < 0.0000), and there was also improvement after guard application (mean VAS score reduction of 0.6, *p* = 0.016). However, while pre-treatment VAS scores were not significantly different between the two treatment groups, post-treatment VAS scores in those treated with clonazepam were significantly lower (*p* = 0.0055). Observations made immediately after the end of treatment (results not shown) indicate a significant reduction in VAS with both treatments. Failure to fully blind this study, as well as other factors, e.g., frequent use of a tongue protector, could significantly affect the assessment of the 4-week treatment.

Significant correlations between psychological and/or clinical variables are shown in Table 4.

A significant covariation was observed between the scores of most psychological and quality of life tests and clinical parameters of xerostomia, nicotinism, and baseline VAS (visual analogue scale) in all subjects with BMS. In addition, significant positive correlations were found between taste disorders, BDI (Beck depression inventory), and AIS (Athenian insomnia scale), as well as a negative one with DOM1. In terms of psychoticism, the Eysenck questionnaire showed only a negative covariation with the BMS type. Except for this psychological parameter, a generally strong covariation was also found between the assessed psychological parameters—depression scale and sleep disorders; quality of life domains and personality dimensions; sleep disorders and DOM1; extroversion/introversion and neuroticism; between the individual quality of life domains; and between the extroversion/introversion and neuroticism. In females, there is strong correlation between the level of depression and all domains of quality of life; in men, this relationship was confirmed only for the somatic domain. Changes in VAS scores in clonazepam-treated patients had a significantly negative correlation with VAS0 R: −0.49 (*p* = 0.07), AIS R: −0.52 (*p* = 0.005), and EPQR N R: −0.47 (*p* = 0.012), as well as a positive one with the HRQoL (health-related quality of life) score in domain 4. Changes in VAS scores in tongue protector users did not significantly correlate with any of the variables assessed.

A multiple regression model emerged, in which changes in VAS values during the assumed follow-up period were most strongly dependent on BMS type and treatment (F value = 35.98, *p* < 0.0000, R = 0.76, R^2^ = 0.571, and adjusted R^2^ = 0.555). Model verification showed: significance of partial regression coefficients (*p* < 0.02), no collinearity between independent variables (tolerance for both variables 0.99, R^2^ = 0.006), scatter plot of residuals against predicted values as a uniform cloud, no autocorrelation of residuals (Durbin–Watson d-statistic= 1.89), normal distribution plot of residuals, and mean of outliers = 0.

Emergent multiple regression model: change in VAS = 5.03 + 0.67 × BMS − 2.68 × treatment type ± 1.275.

## 4. Discussion

BMS is an ongoing multi-specialist challenge. The development of new pathophysiological concepts of BMS offers hope for more effective treatment. However, effective treatment of BMS involving complete remission of all clinical symptoms is still unattainable. Due to the undetermined aetiology of BMS, only symptomatic treatment is possible. The primary aim of the study was to compare the efficacy of two topical treatments for BMS: clonazepam and a tongue protector. The choice of clonazepam was dictated by its best-documented clinical efficacy in this disorder, while the evaluation of the tongue protector resulted from further clinical observations of this form of local treatment. The secondary aim of study considers the impact of the assessment of depression, sleep disorders, and personality types on the clinical symptoms of BMS and quality of life status of patients.

The entire 57-patient group with BMS observed in this study did not differ, with respect to the age and prevalence of taste disorders and xerostomia, as well as the distribution of clinical types of BMS, according to Lamey et al., from the classical description of these clinical changes [3,5,31,36]. However, there is a clear over-representation of males, in relation to the general population with BMS, in which it generally does not exceed 20% [1]. This may be due to the inclusion of individuals with formerly primary BMS, whereas many of the aetiological conditions of previous BMS-related studies were associated specifically with the female sex, e.g., type 2 diabetes, hypothyroidism, anaemias, collagenosis, or chronic oral mucosa mycosis. The over-representation of the male sex in the self-reported study is probably related to the very high percentage of active nicotine users, reaching 49.1%. It is also noticeable that the mean value of pain/burning sensation experienced by our patients was slightly higher than 6 on VAS, whereas, in other prospective studies, it ranged between 3.1 and 5.1 on 10-point scales [5]. This difference may also be a consequence of studying patients with the currently defined BMS only, in which neuropathic pain of peripheral or central origin is arguably more intense than the pain from comorbidities in previous studies and requires the most effective treatment.

The Beck depression inventory confirmed the frequent occurrence of depression in the observed self-reported patients with BMS, as scores corresponding to moderate to severe depression were found in 40.3% of subjects, with a mean test score of 19.6 ± 10.3. Most similar to these results was the observation of de Souza et al. [11], who showed an even slightly higher mean BDI score of 21.4 ± 10.8 in 30 BMS patients. In other studies, BDI results indicated mild depression: −13.1 ± 9 [37] and 11.6 ± 8.3 [38]. In all these studies, significant standard deviations of BDI scores are noted, which was probably the reason no statistically significant difference was found for BDI between BMS and control group in the meta-analyses [12] of two studies [11,38]. A more homogeneous confirmation of BMS association with depression was described in the case of the Hamilton rating scale for depression [12]. In our patients with BMS, up to 54.4% prevalence of sleep disturbance on the Athens insomnia scale was observed. The application of this scale for BMS has not been found in the available literature. However, the association of BMS with daytime sleepiness expressed by other tests, e.g., the Pittsburgh sleep quality index (PSQI) and Epworth sleepiness scale (ESS), has been shown [39]. In a large multicentre case-control study, sleep disturbances have been shown to occur in 79% of BMS cases and consisted of disturbances in sleep onset and duration (insomnia), deterioration in sleep quality, and sleep–wake disorders. Interpretation of the mean scores of the short form of the Eysenck personality questionnaire-revised indicates that individuals with BMS show a moderate tendency towards introversion, neuroticism, and insincerity. Only one attempt to apply the EPQR (concerning the full version) for BMS was found in the literature. However, in the study group of 30 persons, there was a clear predominance of women, a shift towards introversion and neuroticism, as well as, psychoticism, in relation to the control, was also noted [40]. To assess the personality of BMS patients, other psychological tests have been used, e.g., NEO personality inventory, structured clinical interview for diagnostic and statistical manual of mental disorders (DSM-IV), big five inventory, or axis II personality disorders [12]. They confirmed elevated levels of neuroticism, as well as harm avoidance, cancerophobia, and a propensity for obsessive-compulsive disorder. After applying the psychometric Toronto alexithymia scale in a 58-person group with BMS, the occurrence of alexithymia (inability to understand, identify, and express emotions, or ‘emotional illiteracy’) was demonstrated in 79.3% of subjects, in relation to the controls [41], which may pave the way for somatisation symptoms, such as chronic pain and pseudo xerostomia.

In the entire group of our patients with BMS, the deterioration of health-related quality of life could be seen in all the domains of the WHOQoL test. In one of the studies using the same HRQoL test in a group of 58 BMS patients, significant deterioration of QoL, compared to the control group, was found in only the physical and psychological domains [42]. The tools most frequently used in other studies on QoL in BMS were the short form health survey questionnaire (SF-36) for the assessment of HRQoL and OHRQoL oral health impact profile (OHIP-49), or its shorter version, OHIP-14, for the assessment of the oral health-related quality of life. The only meta-analysis to date on the QoL of BMS patients analysing six observational studies has unambiguously confirmed a highly significant deterioration of OHQoL and HRQoL, compared to the control groups [43].

In a correlation analysis of the clinical symptoms and psychological test results of our entire group of BMS patients, close, directly proportional relationships of the intensity of pain/burning sensation and the frequency of xerostomia and dysgeusia with the depression marker used, as well as the intensity of pain/burning sensation and dysgeusia with sleep disorders, intensity of pain/burning sensation, frequency of xerostomia with the level of introversion, intensity of pain/burning sensation, and level of neuroticism, have been found. A significant influence of clinical symptoms on the deterioration of HRQoL can also be observed, especially in the physical, psychological, and social domains. Single observations, concerning the relationships of clinical symptoms of BMS with mental disorders, were also observed in psychometric tests by other authors, although not all of the studies have confirmed them. Al Quaran [44] described very strong correlations of burning sensation intensity with the level of anxiety, depression, and neuroticism. Lee et al. [45] have noted very strong positive correlations of pain interference with anxiety, depression, somatisation, obsessive-compulsive disorders, and sleep quality. Kim et al. [46], in turn, proved in a retrospective study that BMS patients with concomitant anxiety and depression disorders, cancer phobia, and hypochondria experience more intense pain and burning sensation and suffer from dysgeusia and xerostomia more frequently. These interactions demonstrate that mental symptoms, in the form of depression, anxiety, and neuroticism, play the role of accompanying factors in BMS, thus exacerbating its clinical symptoms. This is probably due to the dysregulation of the endocrine and neuroendocrine systems, thus leading to changes in the levels of serum and salivary steroids.

In a bivariate analysis, we have also found very strong correlations of depression with sleep disorders, proneness to introversion and neuroticism, and sleep disorders with introversion and neuroticism. Additionally, in the entire group, higher levels of depression deteriorated the quality of life in all its domains, as well as sleep disorders, deteriorated in the physical domain. We have proven gender differences in the analysis of the effect of depression on the quality of life. In women, this was a correlation with all of the domains, while, in men, there was only a correlation with the physical domain. This different pattern of negative self-observation could be associated with a lower sense of discomfort and higher threshold of emotional reactivity in men. In other observations, a correlation between higher levels of depression and sleep disorders was also observed [13,39,47]. Strong positive correlations of sleep disorders (PSQI) and the VAS score and Hamilton test for depression and anxiety were proven [47]. In a majority of BMS patients, significant positive covariations were confirmed between sleep disorders (PSQI), the level of depression and anxiety in Hamilton tests, and the intensity of pain and general discomfort, as related to the pain and burning sensation [39]. The presence of these covariations could point to a mechanism common to depression, anxiety, sleep disorders, somatization, and chronic pain with dysgeusia, described as BMS, which could be circadian rhythm sleep disorders. Proving chronobiological causes of BMS would certainly provide a new diagnostic and therapeutic impetus for this disease entity. The correlation of higher levels of neuroticism with depression and insomnia and quality of life proven in our study imply that this personality trait can be a predictor significantly affecting the self-esteem and problems reported in the physical function domains, as related to the quality of life, insomnia, and depression, of these patients. Highly neurotic individuals are characterised by unstable emotional reactions, negative self-esteem, tendencies towards excessive self-observation, and low resistance to stressful situations, which predisposes them to the development of psychosomatic disorders secondary to BMS. This has also been indicated by other authors [44,48]. At the same time, it is worth noting that extraversion seems to be a personality trait that stabilises the emotional state of BMS patients, as evidenced by the covariation of its level with depression, insomnia, and the quality of life observed by us, which is in line with the observation of Al Quaran [44].

Following 4 weeks of local administration of clonazepam, three months after the completion of the treatment of BMS, three asymmetrical results were obtained: a decrease in the intensity of pain/burning was observed in 71.4% of patients, there was no improvement in 17.9% of patients (or the severity of pain increased), and 10.7% of patients recovered completely. The entire group experienced a significant decrease in the intensity of pain/burning by, on average, 3.5 on the VAS scale. Table 5 presents a summary of short and long-term observations conducted to date, concerning changes in the intensity of pain/burning in BMS after local or general administration of clonazepam [19,20,21,22,23,24,25,49,50].

In the short-term evaluation, the percentage of complete recoveries ranged from 0 to 15.1% of patients, and the decrease in the intensity of pain on the VAS scale-from 1.8 to 4.9. In the assessment conducted after 10 weeks, the authors observed a complete subsidence of clinical symptoms in 0–33% of patients, as well as a decrease in pain intensity on the VAS scale ranging from 2.2–5.7. Therefore, the results concerning the efficacy of clonazepam use in our patients are within the range obtained in other observations. The highest percentage of complete remissions of BMS symptoms was obtained after general administration of this benzodiazepine [22]; however, the risk of adverse events, e.g., cognitive impairment or addiction, was elevated. Adverse drug reactions were observed in 33% of patients involved in the study, which demonstrated the highest percentage of BMS remissions [22]. In the case of local application, ADRs (adverse drugs reactions) resulting in treatment discontinuation were observed in 6.66% of our patients, as well as 8% and 10% [51] of patients participating in other studies. The ADRs comprised sedation, dizziness, and altered mental status. In a meta-analysis conducted by Cui et al. [52], which involved five studies concerning the application of clonazepam in the treatment of BMS [19,20,21,22,23], a significant decrease in VAS (by 3.72) was observed in a total of 126 patients in the study group and 128 patients in the control group, with a significant heterogeneity of the obtained results. Meta-analytical combination of two studies concerning local application of clonazepam in the treatment of BMS [20,21] also showed a significant reduction in VAS (by 1.5), with maintained homogeneity of the results of these observations [52]. The mechanism of clonazepam’s analgesic effect after its dissolution on the surface of the tongue probably consists in its antagonistic effect on the GABA4-ergic receptors located in the nervous fibres of the oral mucosa [5,51]. These receptors are associated with chloride channels, the opening of which causes hyperpolarization of cell membranes, leading to a decrease in neuronal excitability and inhibition of impulse conduction. However, some BMS patients do not respond to this treatment. It may be related to the central types of BMS, the diagnosis of which is aided by a lack of analgesic effect/subsidence of burning after lingual nerve anaesthesia [53]. Based on the authors’ own two-factor analysis, it was demonstrated that local treatment with clonazepam is less effective in patients who initially experience more severe pain/burning, as well as in the cases of patients experiencing sleep disturbances in the AIS test and those with neurotic type of personality. On the other hand, it was possible to achieve greater reduction in pain/burning intensity in the case of the highest possible level of the quality of life in the domain of the functioning environment in the WHOQoL test. Factors affecting the effectiveness of pharmacological treatment of BMS were also analysed within the framework of two retrospective studies, although the said analyses did not concern local administration of clonazepam. In the analysis of pharmacological treatment of BMS (local/general administration of clonazepam, ALA, tricyclic antidepressants, and capsaicin), Khawaja et al. [54] observed that a significant decrease in pain/burning occurred only for patients in whom the ailments persisted for up to a year and in the case of concomitant hyperlipidaemia, while such an effect did not occur in the case of the concurrent use of neuropathic medications. After general application of clonazepam, Ko et al. [20] observed that there was a greater reduction in pain in the case of patients with higher initial intensity of pain and in those for whom it was combined with dysgeusia and/or xerostomia; additionally, the lack of an effect of this treatment was associated with the occurrence of multiple psychological symptoms registered in the SCL-90-R test, especially interpersonal sensitivity and paranoid ideation. Differences in the relationship between the treatment effects and initial intensity of pain, in comparison with our observations, may stem from the different routes of clonazepam administration and the fact that we only analyzed patients after excluding the known local and general conditions of burning the oral mucosa.

After application of the tongue protector, patient self-evaluation showed a significant reduction in pain/burning intensity in 62% of those treated for primary BMS; the mean reduction in VAS amounted to 0.6 points and was statistically significant. Therefore, a much higher efficacy of treatment was observed in the first group. The originators of this form of BMS treatment observed a decrease of 3.6 in the VAS score, immediately after a two-month-long use of the tongue protector, which was significantly higher than in the control group [30]. In addition, they demonstrated that the improvement in the quality of life related to oral health, monitored with the use of the OHIP49 test, and quality of life related to health in the SF-36 test did not observe a significant impact on the level of anxiety in depression. Differences in the efficacy of BMS treatment, with respect to self-observation, may stem from a shorter period of wearing protectors, as well as long-term assessment of the effects of treatment. In another study conducted by the same authors, three-month-long wearing of the protector did not cause a significant improvement in the distribution of pain, assessed as mild, moderate, and severe, as well as in the changes in the OHRQoL and level of depression [55].

In the multiple regression model, the authors selected independent variables that had a significant impact on reducing pain intensity in BMS in clinical BMS, as well as the type of the conducted treatment. The selected model explains the variation in pain intensity in 57%, with treatment type explaining 70% and BMS-22% of this variation. An individual decrease in pain intensity in the VAS scale is associated with local application of clonazepam by 2.68, with an increase by 0.67 of the clinical number of BMS type.

There were several limitations to the conducted study. First and foremost, the effectiveness of the treatment was only related to the changes in pain intensity, with the most common somatosensory symptoms of BMS, dysgeusia, and xerostomia being ignored. They could also be objectively assessed using electrogustometry and sialometry. Secondly, a major limitation of evaluating the effectiveness of ongoing BMS treatment was the lack of a control group. This was due, in part, to the inability to assess the placebo effect for the second group. Thirdly, also after treatment, health-related quality of life should be assessed, as well as in the case of topical treatment, perhaps even better related to oral health with the OHIP-49 or OHIP-14. Fourthly the limitation is also not presenting the results assessed after 4 weeks of therapy. The study showed reductions in VAS in both groups; however, in our opinion, the lack of complete blindness could have had a significant impact on these results. Fifth, it appears advisable to personalise BMS treatment more by employing diagnostic anaesthesia of the lingual nerve, thus indicating the need for systemic treatment, in the case of the central form of this syndrome. This would be an attempt to reduce the percentage of patients not responding to local treatment. The accepted concept of comparing the two forms of local treatment has, from the outset, favoured significantly longer daily pharmacological treatment over three-hourly protection of the tongue alone from parafunctional trauma and thermal and chemical stimuli.

## 5. Conclusions

In conclusion, burning mouth syndrome, because of its frequent occurrence and diagnostic and therapeutic difficulties, is an ongoing multi-specialist challenge, although primarily for dentists. There is still no gold standard of therapy, in terms of the drug, form of administration, and duration of therapy.

Pain has a strong influence on the overall condition of the patient. In a bivariate analysis, we have found very strong correlations of depression with sleep disorders, proneness to introversion and neuroticism, and sleep disorders with introversion and neuroticism. We confirmed that the disease mainly affects postmenopausal women. Significantly greater improvement in VAS scores was shown in those patient treated with clonazepam.

The development of new pathophysiological concepts of BMS, e.g., chronobiological, offers hope for more effective causal treatment.

Considering the influence of BMS on the quality of life and mental disorders in most patients, and further research on the possibilities of therapy seems to be very important.

## Figures and Tables

**Figure 1 ijerph-19-08999-f001:**
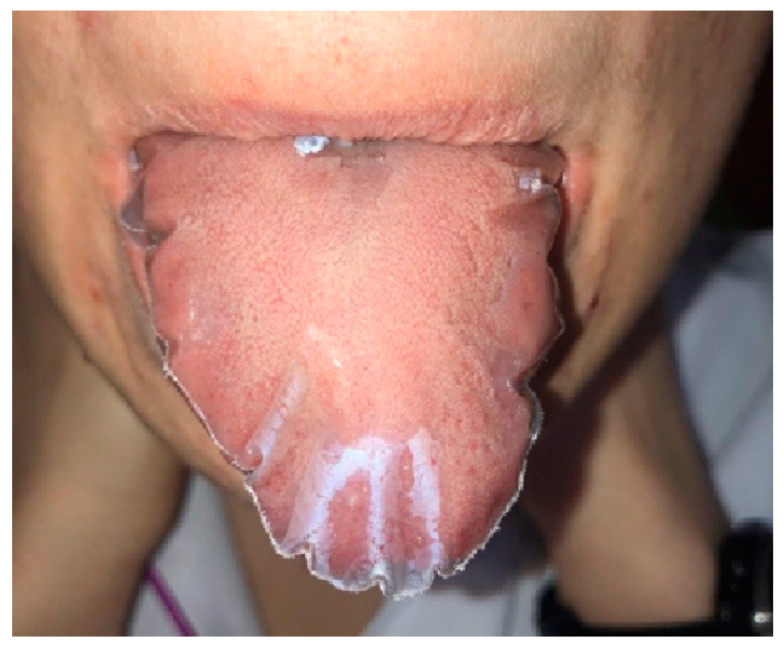
Patient with tongue protector.

**Table 1 ijerph-19-08999-t001:** Treatment protocol.

	1 Week	2 Week	3 Week	4 Week
Group: clonazepam	0.5 mg in the morning once a day	0.5 mg in the morning 2 mg before asleep	0.5 mg in the morning 0.5 mg at midday 2 mg before asleep	0.5 mg in the morning 0.5 mg before asleep
Group: tongue protector	3 times a day (one hour after meals)	3 times a day (one hour after meals)	3 times a day (one hour after meals)	3 times a day (one hour after meals)

**Table 2 ijerph-19-08999-t002:** Descriptive analysis for the whole group (*n* = 57), clonazepam (*n* = 28), and tongue protector (*n* = 29) groups.

Variable	Values
Sex	F 34 (59.6%), M 23 (40.4%)
Age	Mean 62.66 ± 7.6 (52–77)
VAS value	Before treatment 6.07 ± 1.88 (2–9) 3 months after treatment 4.04 ± 2.5 (0–9)
Taste disorders	N-27 (47.4%), Y-30 (52.6%)
Xerostomia	N-34 (59.6%), Y-23 (40.4%)
Smoking status	N-29 (50.9%), Y-28 (49.1%)
Depression (BDI)	Mean 19.61 ± 10.3, values < 20—34 (59.6%), 20–25—5 (8.8%), >25—18 (31.5%) First group-mean 21.3 ± 8.6, second group mean 18.03 ± 11.6
Insomnia symptomatology (AIS)	Mean 9.24 ± 4.1, values < 8—26 (45.6%), ≥8—31 (54.4%) First group-mean 9.32 ± 4.1, second group-mean 9.17 ± 4.3
Quality of life WHO (WHOQOL-BREF)	Means and intervals: DOM1—20.91 ± 2.9 (9–25), DOM2—19.44 ± 3.2 (12–25), DOM3—8.63 ± 2.2 (5–11), DOM4—28.33 ± 3.3 (21–34) Means in the first and second group: DOM1—20.21 ± 2.9 vs. 21.59 ± 2.7, DOM2—18.96 ± 2.5 vs. 19.9 ± 3.8, DOM3—8.07 ± 2.0 vs. 9.17 ± 2.2, DOM4—27.71 ± 2.8 vs. 28.93 ± 3.6
Personality questionnaire acc Eysencka (EPQR-S)	Means: E/I—4.96 ± 3.0, N—6.58 ± 2.9, P—3.96 ± 1.7, L—7.77 ± 1.7 Means in the first and second group: E/I—4.32 ± 2.7 vs. 5.59 ± 3.2, N—7.54 ± 2.6 vs. 5.66 ± 2.8, P—4.07 ± 2.0 vs. 3.86 ± 1.4, L—7.82 ± 1.8 vs. 7.72 ± 1.7

**Table 3 ijerph-19-08999-t003:** BMS treatment outcomes.

Group	Results
Total (*n* = 57)	No improvement VAS—15 (26.3%), improvement VAS—38 (66.7%), healing—4 (7%) VAS before treatment 6.07 ± 1.9, after treatment 4.03 ± 2.5; *p* < 0.0000
First group (*n* = 28)	No improvement VAS—5 (17.9%, without worsening VAS), improvement VAS—20 (71.4%), healing—3 (10.7%) VAS before treatment 6.57 ± 1.5, after treatment 3.07 ± 1.82; *p* < 0.0000
Second group (*n* = 29)	No improvement VAS—10 (34.4%, including 4 with worsening VAS—13.8%), improvement VAS—18 (62.1%), healing—1 (3.4%) VAS before treatment 5.59 ± 2.1 ^1^, after treatment 4.97 ± 2.71 ^2^; *p* = 0.016

^1^ VAS difference between groups before treatment *p* = 0.054; ^2^ difference in VAS values between groups after treatment *p* = 0.0055.

**Table 4 ijerph-19-08999-t004:** Correlations of psychological and clinical variables.

Group	Pychological or Clinical Variable	Demonstrated Significant Correlations with Values Rip
The whole group	BDI	Taste disorders R: 0.37 (*p* = 0.004), xerostomia R: 0.54 (*p* < 0.000), Nicotinism R: 0.73 (*p* < 0.000), VAS1 R: 0.83 (*p* < 0.000) AIS-R: 0.57 (*p* < 0.000), DOM1-R: −0.84 (*p* < 0.000), DOM2-R: −0.64 (*p* < 0.000) DOM3-R: −0.55 (*p* < 0.000), DOM4-R: −0.41 (*p* = 0.001), E/I-R: −0.65 (*p* < 0.000 N-R: 0.65 (*p* < 0.000), L-R: 0.32 (*p* = 0.016)
	AIS	Taste disorders R: 0.45 (*p* < 0.000), nicotinism R: 0.4 (*p* = 0.001), VAS1 R: 0.56 (*p* < 0.000), DOM1-R: −0.51 (*p* < 0.000), E/I-R: −0.34 (*p* = 0.008), N-R: 0.44 (*p* < 0.000)
	DOM1	Taste disorders R: −0.37 (*p* = 0.003), nicotinism R: −0.44 (*p* < 0.000), nicotinism R: −0.6 (*p* < 0.000), VAS1 R: −0.72 (*p* < 0.000), DOM2-R: 0.58 (*p* < 0.0000), DOM3-R: 0.5 (*p* < 0.000), DOM4-R:0.51 (*p* < 0.000), E/I-R: 0.59 (*p* < 0.000), N-R: −0.65 (*p* < 0.000)
	DOM2	Xerostomia R: −0.38 (*p* = 0.003), nicotinism R: −0.42 (*p* = 0.001), VAS1 R: −0.56 (*p* < 0.000), DOM3-R: 0.68 (*p* < 0.000), DOM4-R: 0.5 (*p* < 0.000), E/I-R: 0.61 (*p* < 0.000), N-R: −0.61 (*p* < 0.000)
	DOM3	Xerostomia R: −0.42 (*p* = 0.001), nicotinism R: −0.46 (*p* < 0.000), VAS1 R: −0.46 (*p* < 0.000), DOM4-R: 0.54 (*p* < 0.000), E/I-R: 0.75 (*p* < 0.000), N-R: −0.56 (*p* < 0.000), L-R: −0.34 (*p* = 0.009)
	DOM4	Nicotinism R: −0.41 (*p* = 0.002), VAS1 R: −0.46 (*p* < 0.000), E/I-R: 0.48 (*p* < 0.000), N-R: −0.68 (*p* < 0.000)
	E/I	Xerostomia R: −0.32 (*p* = 0.015), nicotinism R: −0.45 (*p* < 0.000), VAS1 R: −0.54 (*p* < 0.000), N-R: −0.6 (*p* < 0.000), L-R: −0.36 (*p* = 0.006)
	N	Nicotinism R: 0.57 (*p* < 0.000), VAS1 R: 0.58 (*p* < 0.000)
	P	Typ BMS R: −0.39 (*p* = 0.003)
	L	Xerostomia R: 0.32 (*p* = 0.014), nicotinism R: 0.33 (*p* = 0.012)
Female	BDI	DOM1-R: −0.86 (*p* < 0.000), DOM2-R: −0.77 (*p* < 0.000), DOM3-R: −0.66 (*p* < 0.000), DOM4-R: −0.51 (*p* = 0.002)
Male	BDI	DOM1-R: −0.82 (*p* < 0.000)
First group	VAS change following treatment	VAS0 R: −0.49 (*p* = 0.07), AIS R: −0.52 (*p* = 0.005), EPQR N R: −0.47 (*p* = 0.012), DOM4-R: 0.51 (*p* = 0.006)
Second group	VAS change following treatment	None

**Table 5 ijerph-19-08999-t005:** Clinical outcomes of pain intensity reduction in BMS after clonazepam.

Author, Year	Country	Study Design	Age	Numer of Participants	Dailly Dose	Route of Administration	VAS 0	VAS End.	Results < 10 Weeks	Results > 10 Weeks
Woda et al., 1998 [19]	France	Clin-control	62.2 ± 12	25	1–3	topically	6.2	2.6	4 weeks: −3.2	3–29 m: −3.6
Gremeau-Richard, 2004 [20]	France	RCT	65 ± 2	22	3	topically	6.0	3.5	2 weeks: −2.5 W—0, G—13.6%	-
de Rivera Campillo, 2010 [21]	Spain	RCT	64.9 (48–85)	33	0.5–2	topically	7.7	3.0	4 weeks: −4.9 W—15.1%	6 m: −4.7 W—9%
De Castro, 2014 [48]	Brazil	Without control group	59 (30–75)	18	Sol. 0.1 mg/mL × 3	topically	5.6	3.5	2 weeks: −2.1 W—11.1% G—0	-
Kuten-Shorrer et al., 2016 [25]	USA	RCT	60.5	26	Sol. 0.5 mg/mL × 2–4	topically	7.5	1.8	from 3 weeks. to 1 year: −5.7 G—3.8%	
Arduino et al., 2017 [49]	Italy	RCT	65.5 ± 8	15	3	topically	ND	ND	3 weeks: −2.8	3m: −2.2
Amos et al., 2011 [22]	Australia	Clin-control	56.7 ± 2	36	0.5–2.25	System.	6.9	2.3	10 weeks: −3.8	6m: −4.6 W−33.3%
Ko et al., 2012 [24]	Korea	Without control group	58.5 ± 11	100	0.5–1	System.	5.3	3.5	4 weeks: −1.8	-
Heckmann et al., 2012 [23]	Germany	RCT	65 ± 12	10	0.5	System.	7.4	4.5	8 weeks: −2.9	-

RCT—randomised controlled trials; ND—no data; W—healed; G—no effect or worsening; VAS0—at baseline.

## Data Availability

Not applicable.

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
