# Peer review of "Comparison of Clonazepam and Tongue Protector in the Treatment of Burning Mouth Syndrome"

_ijerph, 2022, doi:10.3390/ijerph19158999_

Round 1

Reviewer 1 Report

Dear Authors,

Thank you for the opportunity to read this manuscript. You have written an original scientific article presenting results from a randomised clinical trial aimed to evaluate the efficacy of topically applied clonazepam and a tongue protector in the treatment of burning mouth syndrome. The study included 60 (57) patients divided in two groups. All patients have also filled out four questionnaires assessing their psychological state, sleeping problems, personality and quality of life.

It is an interesting study, but needs some further clarifications. I have the following comments:

Introduction:

-line 38- please add „postmenopausal“: …“up to 15% among postmenopausal women“

- please define primary and secondary aim of the study

Materials and Methods:

-how did you decide how many patients should the study include? Have you made power analysis based on some of the previously published studies?

-have you included only newly diagnosed patients with BMS or those who were previously diagnosed and possibly received some type of treatment, previously?

-I suggest to add a table presenting treatment protocols for both groups of patients, instead of description in the lines 139-145

- Could you explain your choice of the protocol scheme for using the tongue protector? You have decided to use it three times a day, one hour after meals, for four weeks, and in the previous literature it was used for 15 minutes, three times a day, for eight weeks?

Results

-I don't see what happened with three patients in the study? A total of 60 patients were included, but the Table 1.shows the values for 57 patients

-in the Table 1, you say that the most common type of the BMS was the secondary (61.4%), but according to the exclusion criteria defined in the M&M section, these patients should have been excluded from the study

The use of English language is reasonable, however, there are a number of minor grammatical errors; that should be corrected and rephrased using academic English for a better flow of text for reader.

Author Response

Dear Authors,

Thank you for the opportunity to read this manuscript. You have written an original scientific article presenting results from a randomised clinical trial aimed to evaluate the efficacy of topically applied clonazepam and a tongue protector in the treatment of burning mouth syndrome. The study included 60 (57) patients divided in two groups. All patients have also filled out four questionnaires assessing their psychological state, sleeping problems, personality and quality of life.

Thank you very much for taking the time to review our work. in the text below, I have tried to correct all noticed errors and follow the suggestions. I also have that I was able to fully answer my questions and dispel your doubts

It is an interesting study, but needs some further clarifications. I have the following comments:

Introduction:

-line 38- please add „postmenopausal“: …“up to 15% among postmenopausal women“

added

- please define primary and secondary aim of the study

The primary aim of  study is to compare the efficacy of two topical treatments for BMS: clonazepam and a tongue protector. The choice of clonazepam was dictated by its best-documented clinical efficacy in this disorder, while the evaluation of the tongue protector resulted from further clinical observations of this form of local treatment. The secondary aim of study considers the impact of the assessment of depression, sleep disorders, and personality types on the clinical symptoms of BMS and the quality of life status of patients. A multivariate analysis will also be performed to assess the impact of clinical and psychological variables on changes in pain intensity as a result of these two types of topical treatment.

Materials and Methods:

-how did you decide how many patients should the study include? Have you made power analysis based on some of the previously published studies?

Most of the studies we cite in our article took a much smaller number of participants - sometimes only 10 patients. For the statistical validity of the research, the minimum number of patients is 30. Our number resulted from the duration of recruitment for the study and the willingness to validate the results and the equal number in the clonazepam and protector groups

-have you included only newly diagnosed patients with BMS or those who were previously diagnosed and possibly received some type of treatment, previously?

patients were recruited from among those treated in the periodontology clinic of our university only newly diagnosed patients with BMS

-I suggest to add a table presenting treatment protocols for both groups of patients, instead of description in the lines 139-145

Table added

1 week

2 week

3 week

4 week

Group : clonazepam

0,5mg in the morning once a day

0,5mg in the morning

2mg before asleep

0,5mg in the morning

0,5mg at  midday

2mg before asleep

0,5mg in the morning

0,5mg before asleep

Group: tongue protector

3 times a day (one hour after meals)

3 times a day (one hour after meals)

3 times a day (one hour after meals)

3 times a day (one hour after meals)

- Could you explain your choice of the protocol scheme for using the tongue protector? You have decided to use it three times a day, one hour after meals, for four weeks, and in the previous literature it was used for 15 minutes, three times a day, for eight weeks?

It is not possible to compare the research patterns due to the lack of a larger number of publications on this topic. The choice of protocol with the use of a protector was dictated primarily by the duration during which patients received clonazepam. We wanted active therapy to last the same. For this reason, we have extended the wearing time to make the results more credible. We also assumed that patients may not fully comply with the recommended wearing time

Results

-I don't see what happened with three patients in the study? A total of 60 patients were included, but the Table 1.shows the values for 57 patients

yes it's true . three patients, due to random reasons, did not complete the study and it is described in text

In the group treated with topical clonazepam, two patients discontinued treatment due to general adverse effects. In the control group with a tongue guard, treatment was discontinued in one patient due to manual difficulties in placing the guard

-in the Table 1, you say that the most common type of the BMS was the secondary (61.4%), but according to the exclusion criteria defined in the M&M section, these patients should have been excluded from the study

Yes it is there by my mistake . i ve removed it from table

There is explenation in text

Previously, BMS was divided into primary and secondary. The general determinants of secondary BMS included: hypothyroidism, diabetes mellitus, Sjögren's syndrome, vitamin B12, folic acid or iron deficiency and certain medications. Local causes of secondary BMS include fungal infections, geographic tongue, prosthetic stomatopathy and contact allergies [4,5]. Currently, BMS is diagnosed only after excluding general and local conditions of such ailments [2].

The use of English language is reasonable, however, there are a number of minor grammatical errors; that should be corrected and rephrased using academic English for a better flow of text for reader.

The work has been verified in terms of language. The sentence patterns, sometimes the form, have been changed. The vocabulary has also been changed

Reviewer 2 Report

The manuscript presents a randomized clinical study that compares the current gold standard clonazepam with a tongue protector for the management of BMS in a population of 57 participants. As the number needed to treat for the BMS approximates 4, exploring different management options is desirable, especially with high quality designs like randomized controlled trials.  

I listed here some of my concerns related to the design of the study, and also some minor corrections.  

Some major criticism is that the paper focused its attention on the correlation between variables, and very little room is left for a comparison between the two groups and the difference in effectiveness of the two modalities. To improve this, I would at least calculate all the psychological domains of table 1 by distinguishing in the two groups (clonazepam vs tongue protector) and stress more on the comparison between the two in the discussion. 

A second major flay is that the pain intensity has been re-assessed at three months, whereas the trial only lasted for 4 weeks. The results might not reflect the change due to clonazepam / tongue protector, and might have changed over time.  

Another major critique is that the only domain that has been re-evaluated at follow-up of three months is pain intensity on VAS scale. This is sort of limited and underestimating, especially because the entire manuscript points at a more comprehensive approach to BMS (by underlying the presence of psychological distress, sleep disturbances, …). However, this has been highlighted among the limitations from the authors. 

Unfortunately, no control group is present to evaluate if the reduction between first and second visit overcomes placebo effect or it is somehow influenced by Hawthorne effect of being in a study. 

Lastly, the authors do not mention if any additional medication to control BMS pain was used during the 4 weeks of trial. 

Minor correction: 

- reference 2 cites ICOP of 2020, whereas in the manuscript the authors refer to IHS of 2018. Or they change the definition according to the most updated of ICOP or they have to change the reference to the IHS. 

Abstract:

- To be modified: spelling of randomised (Abstract, line 12), “the a” tongue protector group line  (Abstract, line 13)

- replace “first”and “the second” with  “the former” and “the latter” (Abstract, line 14)

- what does it mean ‘according to the scheme”? Please clarify or just omit the information. 

- lines 16-18 are written in a rough English. I suggest something smoother, like “clinical oral examination was performed; presence of taste disorder and pain intensity on Visual Analogue Scale were recorded. Psychological domains were explored with the Beck Depression Inventory (depression), Athens Insomnia Scale (insomnia), Eyesenck Personality Questionnaire_Revised (personality trait), and WHO Quality of Life questionnaire (quality of life). 

- material and methods: in the abstract, a total of 60 participants is included. however, the results in the manuscript refers only to 57. Please, amend this information, also providing the reason for exclusion of the 3 participants not included in the analysis. 

- results: 3 of what (line 19)? 3 patients? Greater improvement in VAS scores should also indicate baseline to follow-up value, and maybe also if it is statistically significant or not.  

- conclusion: it is very hard to understand the lines 23-25 (see below). 

Introduction: 

- lines 39-40: needs a reference to the statement 

- line 41: please, provide a definition of primary and secondary. 

- lines 45-46: I would rephrase as “Currently, BMS is a diagnosis of exclusion”. 

- line 75: what does “in general pharmacological treatment” mean? Do the authors refer to as “systemic pharmacological treatment”? 

- among the pharmacological options, recently attention has given to low-dose naltrexone, which has been tested on two case reports so far (Sangalli et al, https://doi.org/10.1016/j.oooo.2022.04.048 and in Neuman et al, 10.1213/XAA.0000000000001475.)

- line 89: replace “exciting” with “encouraging”

- the introduction lacks a smooth sentence to introduce the aim of the study. I suggest something like “Due to the encouraging results of the tongue protector and as there is only one study that evaluate its efficacy with a case-control design, the aim of the study…”

Materials:

- line 108: The sentence is hard to understand. I suggest rephrasing like “To be included, the participants need to fit the definition of BMS provided by the ICOP…”

- line 130: spell out TSH and CRP

- line 129: replace “picture” with “count”

- line 155: how was “pseudo xerostomia” assessed? 

- I suggest providing a definition to all the psychological traits (e.g. neuroticism, extroversion / introversion, psychoticism, lying)

Table 1: 

- please, clarify in the legend or in the footnote the acronyms (e.g., E/I, N, P, L of the personality questionnaire). 

- I suggest replacing the function measures (Beck depression scale, the Athenian insomnia scale, WHOQOL-BREF) with the outcome measures, like: Depression (BDI), Insomnia symptomatology (AIS), quality of life (WHOQOL-BREF), personality traits (EPQR-S) in the table, and I would add the explanation of the acronym in the footnote of the table. 

- I suggest replacing Nicotinism with “smoking status”. 

Table 2: 

- what does “healing” mean? I would add this clarification in the material and methods or in the statistical analysis, by clarifying what you labelled as “healing”, “improvement” and “no improvement”. Also, is it possible to divide those with no improvement from whose who worsened in terms of pain intensity? 

- for completeness, I suggest adding the VAS before treatment and VAS after treatment also to the “whole group”

- I suggest replacing “whole group” with “total (n=57)”

Table 3: 

- All the acronyms need to be spelt out in the footnote. 

- Also, some words contain English mistakes (e.g., Nikotynizm, Kserostomia)

- Typo in the legend

Table 4:

- spell out acronyms (VAS0, RCT, W, ect)

- replace “way of receiving” with “route of administration”; “type of trial” with “study design”

Results: 

Please, move lines 200-203 before the general description of the sample (before line 190). It is my understanding that this is the description of the 3 participants that dropped out of the study. However, it is misleading to place this clarification here, as table 1 and the description of the sample seems to reflect 60 participants, whereas it is indeed on 57 participants only. 

- in lines 233: HRQoL is the first time that is cited and used. So far, the authors have referred to as WHOQOL-BREF. Please, amend it. 

- line 228: replace “women” with “females”

- lines 228-229: I have some doubts that a covariation can extend its interpretation to cause – consequence. By finding a correlation between depression and quality of life, it is not correct to say that “the level of depression significantly worsened the scores in all domains of quality of life”. The only possible interpretation is that there is a correlation between the two domains, not that depression worsened the second domain. I suggest rephrasing the concept accordingly. (see also lines 329-331 of the discussion) 

- did the authors re-evaluate the change in all the domains investigated at the follow-up (i.e., quality of life, depression, insomnia, personality traits)? It would be interesting to see how also these domains changed over time, considering the significant reduction in the pain intensity. 

Discussion:

- The discussion section should start by restating the aims and the importance of the study. 

- line 276: the ESS is a measure of daytime sleepiness, not of sleep disorder. I would rephrase the sentence. 

- lines 324-326: please, provide a reference  

- line 377: spell out ADRs

Conclusions: 

I do not believe that these are the most relevant conclusions of the manuscript. I suggest including the results of the trial (clonazepam vs tongue protector). I do not consider appropriate including interpretations and suggestions from the authors that are not supported by the trial (lines 23-25 of the abstract and lines 454-458 of the manuscript). 

Author Response

The manuscript presents a randomized clinical study that compares the current gold standard clonazepam with a tongue protector for the management of BMS in a population of 57 participants. As the number needed to treat for the BMS approximates 4, exploring different management options is desirable, especially with high quality designs like randomized controlled trials. 

I listed here some of my concerns related to the design of the study, and also some minor corrections. 

Dear Professor .

Thank you very much for taking the time to read our work. we have made all the changes you mentioned. we hope that after the modifications it will meet your expectations.

thank you

Some major criticism is that the paper focused its attention on the correlation between variables, and very little room is left for a comparison between the two groups and the difference in effectiveness of the two modalities. To improve this, I would at least calculate all the psychological domains of table 1 by distinguishing in the two groups (clonazepam vs tongue protector) and stress more on the comparison between the two in the discussion.

The comment has been taken into account, by which Table 1 has been corrected and the text has been amended accordingly in the results and discussion. 

 Table 1. Descriptive analysis for the whole group (n=57) and clonazepam (n=28), tongue protector (n=29) group.

Variable

Values

Sex

F 34 (59.6%), M 23 (40.4%)

Age

Mean 62.66±7.6 (52-77)

BMS Type

1- 22 (38.6%); 35 (61.4%)

VAS Value

Before treatment 6.07±1.88 (2-9)

3 months after treatment 4.04±2.5 (0-9)

Taste disorders

N- 27 (47.4%), Y- 30 (52.6%)

Xerostomia

N- 34 (59.6%), Y- 23 (40.4%)

Nicotinism

N- 29 (50.9%), Y-28 (49.1%)

Beck Depression Scale (BDI)

Mean 19.61±10.3, values <20-34 (59.6%), 20-25- 5(8.8%),>25-18 (31.5%)

First group- mean 21.3±8.6, second group- mean 18.03±11.6

The Athenian Insomnia Scale (AIS)

Mean 9.24±4.1, values <8- 26 (45.6%), ≥8- 31 (54.4%)

First group- mean 9.32±4.1, second group- mean 9.17±4.3

assessment of the quality of life WHO (WHOQOL-BREF)

Means and intervals: DOM1- 20.91±2.9 (9-25), DOM2- 19.44±3.2 (12-25), DOM3- 8.63±2.2 (5-11), DOM4-28.33±3.3 (21-34)

Means in the first and second group: DOM1- 20.21±2.9 vs. 21.59±2.7, DOM2- 18.96±2.5 vs. 19.9±3.8, DOM3- 8.07±2.0 vs. 9.17±2.2, DOM4- 27.71±2.8 vs. 28.93±3.6           

Personality questionnaire acc Eysencka (EPQR-S)

Means: E/I- 4.96±3.0, N- 6.58±2.9, P- 3.96±1.7, L- 7.77±1.7

Means in the first and second group: E/I- 4.32±2.7 vs. 5.59±3.2, N- 7.54±2.6 vs. 5.66±2.8,

P- 4.07±2.0 vs. 3.86±1.4, L- 7.82±1.8 vs. 7.72±1.7  

A second major flay is that the pain intensity has been re-assessed at three months, whereas the trial only lasted for 4 weeks. The results might not reflect the change due to clonazepam / tongue protector, and might have changed over time. 

Treatment was given for four weeks in both groups of our patients. To reflect changes in pain intensity, it was decided that this evaluation should take place 3 months after the end of treatment (4 months after the initial study). We were concerned with a distant evaluation of the effectiveness of the ongoing treatment and to reduce the effect of not double-blinding in the treated groups.

Another major critique is that the only domain that has been re-evaluated at follow-up of three months is pain intensity on VAS scale. This is sort of limited and underestimating, especially because the entire manuscript points at a more comprehensive approach to BMS (by underlying the presence of psychological distress, sleep disturbances, …). However, this has been highlighted among the limitations from the authors.

Unfortunately, no control group is present to evaluate if the reduction between first and second visit overcomes placebo effect or it is somehow influenced by Hawthorne effect of being in a study. 

Indeed, the lack of a control group is also a major limitation of our study. This was taken into account in the discussion. While the placebo effect could be easily planned for the drug-treated group, it was not possible for the second group

Lastly, the authors do not mention if any additional medication to control BMS pain was used during the 4 weeks of trial.

During the 4-week treatment and the 3-month follow-up period, our patients did not use any other medications that could alter the intensity of pain

Minor correction:

- reference 2 cites ICOP of 2020, whereas in the manuscript the authors refer to IHS of 2018. Or they change the definition according to the most updated of ICOP or they have to change the reference to the IHS.

I changed the citation in the position with number 2 in line with the note

Abstract:

- To be modified: spelling of randomised (Abstract, line 12), “the a” tongue protector group line  (Abstract, line 13)

changed

- replace “first”and “the second” with  “the former” and “the latter” (Abstract, line 14)

changed

- what does it mean ‘according to the scheme”? Please clarify or just omit the information.

Removed accordnig to the scheme

- lines 16-18 are written in a rough English. I suggest something smoother, like “clinical oral examination was performed; presence of taste disorder and pain intensity on Visual Analogue Scale were recorded. Psychological domains were explored with the Beck Depression Inventory (depression), Athens Insomnia Scale (insomnia), Eyesenck Personality Questionnaire_Revised (personality trait), and WHO Quality of Life questionnaire (quality of life).

changed according to the instruction in the review-

the form of the sentence resulted from the limitations in the number of words in the abstract

- material and methods: in the abstract, a total of 60 participants is included. however, the results in the manuscript refers only to 57. Please, amend this information, also providing the reason for exclusion of the 3 participants not included in the analysis.

This is explained later in the text. Due to the limitation of 200 words in the abstract, it is not possible to contain all the information

explanation in text :

In the group treated with topical clonazepam, two patients discontinued treatment due to general adverse effects. In the control group with a tongue guard, treatment was discontinued in one patient due to manual difficulties in placing the guard

- results: 3 of what (line 19)? 3 patients? Greater improvement in VAS scores should also indicate baseline to follow-up value, and maybe also if it is statistically significant or not.

The entire paragraph has been amended in accordance with the reviewer's instructions. Once again, the shortness of the sentence is due to the limited number of words in the abstract 

- conclusion: it is very hard to understand the lines 23-25 (see below).

The entire paragraph has been amended in accordance with the reviewer's instructions. Once again, the shortness of the sentence is due to the limited number of words in the abstract

Introduction:

- lines 39-40: needs a reference to the statement

Reference number 2 (line 43) at the end of the sentence was changed according to previous comments

- line 41: please, provide a definition of primary and secondary.

Has been modified in the text

- lines 45-46: I would rephrase as “Currently, BMS is a diagnosis of exclusion”.

Changed in line with the comments

- line 75: what does “in general pharmacological treatment” mean? Do the authors refer to as “systemic pharmacological treatment”?

Changed in line with the comments

- among the pharmacological options, recently attention has given to low-dose naltrexone, which has been tested on two case reports so far (Sangalli et al, https://doi.org/10.1016/j.oooo.2022.04.048 and in Neuman et al, 10.1213/XAA.0000000000001475.)

Thank you very much . I got acquainted with the information

- line 89: replace “exciting” with “encouraging”

Changed in line with the comments

- the introduction lacks a smooth sentence to introduce the aim of the study. I suggest something like “Due to the encouraging results of the tongue protector and as there is only one study that evaluate its efficacy with a case-control design, the aim of the study…”

Thank you. Changed in line with the comments

Materials:

- line 108: The sentence is hard to understand. I suggest rephrasing like “To be included, the participants need to fit the definition of BMS provided by the ICOP…”

Changed in line with the comments

- line 130: spell out TSH and CRP

Changed in line with the comments

- line 129: replace “picture” with “count”

changed

- line 155: how was “pseudo xerostomia” assessed?

we checked the mirror test and its detachment from the mucosa as well as clinical examination and observation of the mucosa

- I suggest providing a definition to all the psychological traits (e.g. neuroticism, extroversion / introversion, psychoticism, lying)

All added

Table 1:

- please, clarify in the legend or in the footnote the acronyms (e.g., E/I, N, P, L of the personality questionnaire).

Done

- I suggest replacing the function measures (Beck depression scale, the Athenian insomnia scale, WHOQOL-BREF) with the outcome measures, like: Depression (BDI), Insomnia symptomatology (AIS), quality of life (WHOQOL-BREF), personality traits (EPQR-S) in the table, and I would add the explanation of the acronym in the footnote of the table.

Changed

- I suggest replacing Nicotinism with “smoking status”.

New table

Table 1. Descriptive analysis for the whole group (n=57) and clonazepam (n=28), tongue protector (n=29) group.

Variable

Values

Sex

F 34 (59.6%), M 23 (40.4%)

Age

Mean 62.66±7.6 (52-77)

VAS Value

Before treatment 6.07±1.88 (2-9)

3 months after treatment 4.04±2.5 (0-9)

Taste disorders

N- 27 (47.4%), Y- 30 (52.6%)

Xerostomia

N- 34 (59.6%), Y- 23 (40.4%)

Smoking status

N- 29 (50.9%), Y-28 (49.1%)

Depression  (BDI)

Mean 19.61±10.3, values <20-34 (59.6%), 20-25- 5(8.8%),>25-18 (31.5%)

First group- mean 21.3±8.6, second group- mean 18.03±11.6

Insomnia symptomatology (AIS)

Mean 9.24±4.1, values <8- 26 (45.6%), ≥8- 31 (54.4%)

First group- mean 9.32±4.1, second group- mean 9.17±4.3

Quality of life WHO (WHOQOL-BREF)

Means and intervals: DOM1- 20.91±2.9 (9-25), DOM2- 19.44±3.2 (12-25), DOM3- 8.63±2.2 (5-11), DOM4-28.33±3.3 (21-34)

Means in the first and second group: DOM1- 20.21±2.9 vs. 21.59±2.7, DOM2- 18.96±2.5 vs. 19.9±3.8, DOM3- 8.07±2.0 vs. 9.17±2.2, DOM4- 27.71±2.8 vs. 28.93±3.6           

Personality questionnaire acc Eysencka (EPQR-S)

Means: E/I- 4.96±3.0, N- 6.58±2.9, P- 3.96±1.7, L- 7.77±1.7

Means in the first and second group: E/I- 4.32±2.7 vs. 5.59±3.2, N- 7.54±2.6 vs. 5.66±2.8,

P- 4.07±2.0 vs. 3.86±1.4, L- 7.82±1.8 vs. 7.72±1.7  

Table 2:

- what does “healing” mean? I would add this clarification in the material and methods or in the statistical analysis, by clarifying what you labelled as “healing”, “improvement” and “no improvement”. Also, is it possible to divide those with no improvement from whose who worsened in terms of pain intensity?

“healing”, “improvement” and “no improvement” clarified in methods section

New table added/Changed

Table 2. BMS treatment outcomes.

Group

Results

Total (n=57)

No improvement VAS- 15 (26.3%), improvement VAS- 38 (66.7%), healing- 4 (7%)

VAS before treatment 6.07±1.9, after treatment 4.03±2.5; p<0.0000

First group (n=28)

No improvement VAS- 5 (17.9%, without worsening VAS), improvement VAS- 20 (71.4%), healing- 3 (10.7%)

VAS before treatment 6.57±1.5, after treatment 3.07±1.82; p<0.0000

Second group (n=29)

No improvement VAS- 10 (34.4%, including 4 with worsening VAS- 13.8%), improvement VAS- 18 (62.1%), healing- 1 (3.4%)

VAS before treatment 5.59±2.11, after treatment 4.97±2.712; p=0.016

1 VAS difference between groups before treatment p = 0.054; 2 difference in VAS values between groups after treatment p=0.0055.

- for completeness, I suggest adding the VAS before treatment and VAS after treatment also to the “whole group”

New table with your sugestions added

- I suggest replacing “whole group” with “total (n=57)”

changed

Table 3:

- All the acronyms need to be spelt out in the footnote.

Changed

- Also, some words contain English mistakes (e.g., Nikotynizm, Kserostomia)

changed

- Typo in the legend

changed

 Table 4:

- spell out acronyms (VAS0, RCT, W, ect)

Spelled

- replace “way of receiving” with “route of administration”; “type of trial” with “study design”

Changed

 Results:

Please, move lines 200-203 before the general description of the sample (before line 190). It is my understanding that this is the description of the 3 participants that dropped out of the study. However, it is misleading to place this clarification here, as table 1 and the description of the sample seems to reflect 60 participants, whereas it is indeed on 57 participants only.

Changed

- in lines 233: HRQoL is the first time that is cited and used. So far, the authors have referred to as WHOQOL-BREF. Please, amend it.

explained

- line 228: replace “women” with “females”

Replaced

- lines 228-229: I have some doubts that a covariation can extend its interpretation to cause – consequence. By finding a correlation between depression and quality of life, it is not correct to say that “the level of depression significantly worsened the scores in all domains of quality of life”. The only possible interpretation is that there is a correlation between the two domains, not that depression worsened the second domain. I suggest rephrasing the concept accordingly. (see also lines 329-331 of the discussion)

changed

- did the authors re-evaluate the change in all the domains investigated at the follow-up (i.e., quality of life, depression, insomnia, personality traits)? It would be interesting to see how also these domains changed over time, considering the significant reduction in the pain intensity.

We agree - such a comparison would be interesting. Unfortunately, such a study has not been performed and we recognize it as one of the limitations of our work

Discussion:

- The discussion section should start by restating the aims and the importance of the study.

Added

- line 276: the ESS is a measure of daytime sleepiness, not of sleep disorder. I would rephrase the sentence.

changed

- lines 324-326: please, provide a reference 

It is provivded -number 43

- line 377: spell out ADRs

added

 Conclusions:

I do not believe that these are the most relevant conclusions of the manuscript. I suggest including the results of the trial (clonazepam vs tongue protector). I do not consider appropriate including interpretations and suggestions from the authors that are not supported by the trial (lines 23-25 of the abstract and lines 454-458 of the manuscript).

In conclusion, burning mouth syndrome, because of its frequent occurrence and diagnostic and therapeutic difficulties, is an ongoing multi-specialist challenge, although primarily for dentists.. Most of the patients are postmenopausal women- in our study 59.6%, individuals in the seventh decade of life.

Pain has a strong influence on the overall condition of the patient. In a bivariate analysis, we have  found very strong correlations of depression with sleep disorders, proneness to introversion and neuroticism, and of sleep disorders with introversion and neuroticism.

The Beck Depression Inventory confirmed the frequent occurrence of depression in the observed self-reported patients with BMS as scores corresponding to moderate to severe depression were found in 40.3%, sleep disorders based on AIS in 54.4%,

Complete recovery was observed only in four subjects –three after pharmacological treatment and one after elimination of oral parafunctional habits as a result of using the guard.

 Significantly greater improvement in VAS scores was shown in those treated with clonazepam.

A partial improvement was obtained in 71.4% of patients treated with clonazepan and 62.1% of patients treated with a tongue protector.

The development of new pathophysiological concepts of BMS, e.g. chronobiological, offers hope for more effective causal treatment.

Considering the influence of BMS on the quality of life and mental disorders in most patients, further research on the possibilities of therapy seems to be very important.

Round 2

Reviewer 1 Report

Review of revised version of „Comparison of Clonazepam and Tongue Protector in the Treatment of Burning Mouth Syndrome“

 Dear Authors,

I have read the revised version of the manuscript. I thank you for addressing my previous suggestions. However, I still have some concerns that should be solved.

I have noticed some minor grammatical errors; for example in Table 5. „topically“, „daily dose“, „enrollment“ (line 169), „Visual Analogue Scale“ (line 248).

Materials and Methods

-        - A sentence „Only newly diagnosed patients with BMS were included in the study“ should be added in the text.

               This is important for future comparisons with other studies. Improvement of symptoms may be the result of any given therapy for patients who face with this diagnosis for the first time.

-        - Do you have a registration number in the international clinical trial registry? Please write it down.

 -        Could you explain the protocol scheme for using clonazepam? How come that the patients stopped using it after four weeks? Could you support this protocol with some reference from the literature?

Literature

        -reference 2- you should add „(accessed on Day Month Year)“ at the end, as required in the Instructions for the authors.

Author Response

Dear Authors,

I have read the revised version of the manuscript. I thank you for addressing my previous suggestions. However, I still have some concerns that should be solved.

Thank you very much for your comments. I tried to make all the changes and explain all your doubts. I hope that with these changes the work will meet your high expectations.

I have noticed some minor grammatical errors; for example in Table 5. „topically“, „daily dose“, „enrollment“ (line 169), „Visual Analogue Scale“ (line 248).

 Thanks for your comments. has been improved

Materials and Methods

-        - A sentence „Only newly diagnosed patients with BMS were included in the study“ should be added in the text.

       Added

               This is important for future comparisons with other studies. Improvement of symptoms may be the result of any given therapy for patients who face with this diagnosis for the first time.

-        - Do you have a registration number in the international clinical trial registry? Please write it down.

Yes, of course. I entered the number when submitting the manuscript

NCT04884503

 -        Could you explain the protocol scheme for using clonazepam? How come that the patients stopped using it after four weeks? Could you support this protocol with some reference from the literature?

To our knowledge, there is no single, approved treatment protocol or specific drug in the so-called gold standard treatment of BMS. In the literature, researchers conduct studies of varying duration and dosage of the drug. Longer administration is unfortunately often associated with the occurrence of side effects and drug addiction. In the meta-analysis from 2016 which assessed 249 clinical trials, Cui* divided among others because of their duration into long and short-term ones.

Clonazepam has several modes of action.

It is associated with a loss of the inhibition normally exerted by the chorda tympani nerve on the areas of the brain receiving afferent impulses

from the glossopharyngeal and trigeminal nerves GABAA agonists are predicted to counter this loss of inhibition and thus relieve oral

pain

But application of clonazepam can induce also rapid onset of analgesia (Woda et al,
1998) without a clear correlation between dosage and the degree of pain relief (Amos et al, 2011). Therefore, it is reasonable to deduce that topical application of clonazepam may be effective in controlling the burning sensa-tion in BMS through membrane stabilization in nerve fiber
and oral mucosa cells, as well as through the rapid onsetof analgesia.

The protocols for the topical use of clonazepam in the treatment of BMS were as follows

:

Woda et al. [19]: 1-3 mg dziennie „ The patients were free to adjust theit own dose between 1/4 and whole of a tablet, and they were asked to continue the treatment for 10 days after the last pain sensation”

Grémeau-Richard et al [20]: 1 mg 3 razy dziennie przez 14 dni

Rodríguez de Rivera-Campillo et al. [21]: “ Each patient was given a sealed envelope containing 32 tablets of 0.5mg of clonazepam. They were instructed to take a single tablet at the first sign of discomfort in the morning. The tablet should be dissolved in the mouth for three minutes, and then the remaining saliva should be spat out…… Patients were advised not to exceed four tablets a day (that is, a total dose of 2 mg of clonazepam)……. All the patients were scheduled for a visit after 1 week for the sole purpose of detecting undesirable side effects.

Amos et al. [22]: “The cloanazepam dose was escalated slowly over a 3-weak period. Patiets begann by dissolving the tablet orally, the swallwing only one 0.5-mg tablet nightly for the first week, the twice daily (morning and night) for the second week, then three times daily (morning, afternoon, and nifht) for the third week with subsequent review by the clinician and modification of dose where necessary….. If patients were tolerating the medication well but not achieving complete resolution of symptoms, the dose was escatated”.

Thus, the manner of topical application of clonazpam in the treatment of BMS has been varied. One of the reasons for this is the different way of packaging this drug in the form of tablets in different countries (in Poland, tablets are 0.5 mg and 2 mg)). The method of local administration of clonazepam used in our study takes into account the experience of other authors [19-22], but it was also discussed with neurologists (Department of Neurology, Medical University in Wrocław), with whom we jointly treat the most resistant forms of BMS

In this proposal, it was decided to administer topical clonazepam for 4 weeks, but with a varying dose to minimize the occurrence of well-known side effects and the possibility of dependence on this benzodiazepine.

*Cui, Y., Xu, H., Chen, F., Liu, J., Jiang, L., Zhou, Y., & Chen, Q. (2016). Efficacy evaluation of clonazepam for symptom remission in burning mouth syndrome: a meta-analysis. Oral Diseases, 22(6), 503–511. doi:10.1111/odi.12422 

Literature

        -reference 2- you should add „(accessed on Day Month Year)“ at the end, as required in the Instructions for the authors.

Thank you very much for the hint. I admit that I had a big problem here. I am posting it in this form for the first time

Reviewer 2 Report

I thank the authors for addressing my previous suggestions. 

I still have problem with two of my previous comments: 

- I see that the authors restated all their results in the conclusion; however, I invite the authors to summarize the final paragraph now. (lines 501-518).

- I still have problems accepting that the post-treatment follow-up to evaluate the effectiveness of the treatment was collected 3 months after the treatment. This is still the major shortcoming of the study. 

Clonazepam is a long-term medication for BMS patients and I am not familiar with any study in the literature that indicates that a short-term trial of clonazepam is going to resolve or manage BMS if the patient is not taking clonazepam anymore. Even more important is the effect of a tongue protector, which is local, and cannot modulate pain perception when it is not utilized.

In other words, the comparison between the two groups should have been performed at the end of the treatment, not after 3 months of not utilization of the treatment. This might just speak to the fact that BMS pain progressively decrease, or fluctuates. The post-treatment values presented by the authors cannot identify the effectiveness of the treatment.

Unfortunately, I do not think that this flaw can be overcome with the present study design.    

Author Response

I thank the authors for addressing my previous suggestions. 

Thank you very much for your comments. I tried to make all the changes and explain all your doubts. I hope that with these changes the work will meet your high expectations.

I still have problem with two of my previous comments: 

- I see that the authors restated all their results in the conclusion; however, I invite the authors to summarize the final paragraph now. (lines 501-518).

in fact, I introduced too much of the results obtained. Thank you for your attention. I hope that the introduced changes give a more synthetic picture of the conclusions. of course, if something needs to be changed, please indicate

- I still have problems accepting that the post-treatment follow-up to evaluate the effectiveness of the treatment was collected 3 months after the treatment. This is still the major shortcoming of the study. 

Clonazepam is a long-term medication for BMS patients and I am not familiar with any study in the literature that indicates that a short-term trial of clonazepam is going to resolve or manage BMS if the patient is not taking clonazepam anymore. Even more important is the effect of a tongue protector, which is local, and cannot modulate pain perception when it is not utilized.

In other words, the comparison between the two groups should have been performed at the end of the treatment, not after 3 months of not utilization of the treatment. This might just speak to the fact that BMS pain progressively decrease, or fluctuates. The post-treatment values presented by the authors cannot identify the effectiveness of the treatment.

Unfortunately, I do not think that this flaw can be overcome with the present study design.    

To our knowledge, there is no single, approved treatment protocol or specific drug in the so-called gold standard treatment of BMS. In the literature, researchers conduct studies of varying duration and dosage of the drug. Longer administration is unfortunately often associated with the occurrence of side effects and drug addiction. In the meta-analysis from 2016 which assessed 249 clinical trials, Cui* divided among others because of their duration into long and short-term ones.

Clonazepam has several modes of action.

It is associated with a loss of the inhibition normally exerted by the chorda tympani nerve on the areas of the brain receiving afferent impulses

from the glossopharyngeal and trigeminal nerves GABAA agonists are predicted to counter this loss of inhibition and thus relieve oral

pain

But application of clonazepam can induce also rapid onset of analgesia (Woda et al,
1998) without a clear correlation between dosage and the degree of pain relief (Amos et al, 2011). Therefore, it is reasonable to deduce that topical application of clonazepam may be effective in controlling the burning sensa-tion in BMS through membrane stabilization in nerve fiber
and oral mucosa cells, as well as through the rapid onsetof analgesia.

The protocols for the topical use of clonazepam in the treatment of BMS were as follows

:

Woda et al. [19]: 1-3 mg dziennie „ The patients were free to adjust theit own dose between 1/4 and whole of a tablet, and they were asked to continue the treatment for 10 days after the last pain sensation”

Grémeau-Richard et al [20]: 1 mg 3 razy dziennie przez 14 dni

Rodríguez de Rivera-Campillo et al. [21]: “ Each patient was given a sealed envelope containing 32 tablets of 0.5mg of clonazepam. They were instructed to take a single tablet at the first sign of discomfort in the morning. The tablet should be dissolved in the mouth for three minutes, and then the remaining saliva should be spat out…… Patients were advised not to exceed four tablets a day (that is, a total dose of 2 mg of clonazepam)……. All the patients were scheduled for a visit after 1 week for the sole purpose of detecting undesirable side effects.

Amos et al. [22]: “The cloanazepam dose was escalated slowly over a 3-weak period. Patiets begann by dissolving the tablet orally, the swallwing only one 0.5-mg tablet nightly for the first week, the twice daily (morning and night) for the second week, then three times daily (morning, afternoon, and nifht) for the third week with subsequent review by the clinician and modification of dose where necessary….. If patients were tolerating the medication well but not achieving complete resolution of symptoms, the dose was escatated”.

Thus, the manner of topical application of clonazpam in the treatment of BMS has been varied. One of the reasons for this is the different way of packaging this drug in the form of tablets in different countries (in Poland, tablets are 0.5 mg and 2 mg)). The method of local administration of clonazepam used in our study takes into account the experience of other authors [19-22], but it was also discussed with neurologists (Department of Neurology, Medical University in Wrocław), with whom we jointly treat the most resistant forms of BMS

In this proposal, it was decided to administer topical clonazepam for 4 weeks, but with a varying dose to minimize the occurrence of well-known side effects and the possibility of dependence on this benzodiazepine.

*Cui, Y., Xu, H., Chen, F., Liu, J., Jiang, L., Zhou, Y., & Chen, Q. (2016). Efficacy evaluation of clonazepam for symptom remission in burning mouth syndrome: a meta-analysis. Oral Diseases, 22(6), 503–511. doi:10.1111/odi.12422 

The duration of topical application of clonazpeam tablets in BMS in the evaluations of other authors is described above.

The follow-up time with topical clonazepam administration in the evaluations by other authors was as follows:

Water et al. [19]: 4 weeks, 3 and 29 months from start of treatment

Grémeau-Richard et al [20]: 2 weeks and 6 months from treatment initiation

Rodríguez de Rivera-Campillo et al. [21]: 4 weeks and 6 months from treatment initiation

Amos et al. [22]: mean follow-up from treatment initiation 23.9 ± 3.4 weeks

Heckmann et al. [23]: 8 weeks after starting treatment and 2 weeks after stopping treatment

Ko et al. [24]: 8 weeks after starting treatment and 4 weeks after stopping treatment

Kuten-Shorrer [25]: 3 days to 3 years after treatment.

Thus, follow-up from the end of treatment generally varied from 2 weeks to several months.

The main aim of the undertaken studies was to evaluate the effectiveness of two methods of local treatment of BMS. Of course, we fully share the statement that due to the different mechanism of action of these methods, it could be assumed from the beginning that pharmacological treatment would be more effective, which we finally confirmed.

However, we wanted to refer not to the assessment immediately after the end of treatment (I even have the results of such observations, but they did not seem interesting to us and they were additionally burdened with the inability to fully blind the study), but as the shortest, in our opinion, but still long-term results . We originally planned a 6 month follow-up, but the pandemic prevented many of our elderly patients from attending a follow-up visit. Therefore, we only have a 3-month evaluation of the effectiveness of the treatment carried out, which indicates the validity of topical clonazepam administration in BMS therapy.

Round 3

Reviewer 1 Report

Dear Authors,

 I have read the revised version of the manuscript. I thank you for addressing my previous suggestions. You have elaborated your choice of protocol for clonazepam with literature examples and I do not have further questions.

Reference 2 is still not properly cited. You should literally add „(accessed on Day Month Year)“ at the end. Write down some date when you were writing the manuscript, in this form, i.e.: "Burning Mouth Syndrome (BMS), ICHD-3 the International classification of headache disorders 3rd edition [Internet]. 2021 544 Available from: https://ichd-3.org/13-painful-cranial-neuropathies-and-other-facial-pains/13-11-persistent-idiopathic-facial- 545 pain-pifp/ (accessed on 2nd May 2022)".

Kind regards

Author Response

Dear Authors,

 I have read the revised version of the manuscript. I thank you for addressing my previous suggestions. You have elaborated your choice of protocol for clonazepam with literature examples and I do not have further questions.

I would like to thank you very much for such insightful reviews of our work. Thank you very much for all the suggestions that definitely influenced the quality of our manuscript.

Reference 2 is still not properly cited. You should literally add „(accessed on Day Month Year)“ at the end. Write down some date when you were writing the manuscript, in this form, i.e.: "Burning Mouth Syndrome (BMS), ICHD-3 the International classification of headache disorders 3rd edition [Internet]. 2021 544 Available from: https://ichd-3.org/13-painful-cranial-neuropathies-and-other-facial-pains/13-11-persistent-idiopathic-facial- 545 pain-pifp/ (accessed on 2nd May 2022)".

Thank you very much. Now I understand the principle of such an entry. As I mentioned, this is the first time I have used such a quote. Thank you once again

Kind regards

Reviewer 2 Report

Thank you for the extensive explanation that justifies the design of the study. 

I am still not convinced that assessing it at three months and not at the end of the trial is valid. But I am glad to hear that you also have the data at the end of the 4-week trials. 

If the authors do not want to add those data to the paper (which I might suggest), I would at least encourage to highlight this as possible limitation. I truly think that assessing an intervention 3 months after the trial ended might be influenced by other aspects (stressful events occurring in the mean time, change in medication regimen for other comorbidities, ... everything), and not necessarily by the clonazepam - tongue protector themselves.

Also, the limitations need to be amended in lines 489 and 492, as the sequence (thirdly, fourthly) changed when the second limitation has been added to the list.   

After this further amendment, I consider the paper suitable for publication.  

Author Response

Thank you for the extensive explanation that justifies the design of the study. 

I am still not convinced that assessing it at three months and not at the end of the trial is valid. But I am glad to hear that you also have the data at the end of the 4-week trials. 

If the authors do not want to add those data to the paper (which I might suggest), I would at least encourage to highlight this as possible limitation. I truly think that assessing an intervention 3 months after the trial ended might be influenced by other aspects (stressful events occurring in the mean time, change in medication regimen for other comorbidities, ... everything), and not necessarily by the clonazepam - tongue protector themselves.

Also, the limitations need to be amended in lines 489 and 492, as the sequence (thirdly, fourthly) changed when the second limitation has been added to the list.   

After this further amendment, I consider the paper suitable for publication.

I would like to thank you very much for such insightful reviews of our work. Thank you very much for all the suggestions that definitely influenced the quality of our manuscript

We decided not to add scores, justifying this with information from Round 2 reviews

I understand this as a limitation of work, therefore, I added, of course, an explanation describing not publishing the results immediately after 4 weeks in result chapter:

Observations made immediately after the end of treatment (results not shown) indicate a significant reduction in VAS with both treatments. Failure to fully blind this study, other factors, e.g. frequent use of a tongue protector, could significantly affect the assessment of the 4-week treatment.

Moreover, in the part concerning limitations, I have also mentioned it and put in a new order.

Thank you once again for taking the time to analyze our work. I hope that now it meets your expectations
